# TNF-α Plus IL-1β Induces Opposite Regulation of Cx43 Hemichannels and Gap Junctions in Mesangial Cells through a RhoA/ROCK-Dependent Pathway

**DOI:** 10.3390/ijms231710097

**Published:** 2022-09-03

**Authors:** Claudia M. Lucero, Lucas Marambio-Ruiz, Javiera Balmazabal, Juan Prieto-Villalobos, Marcelo León, Paola Fernández, Juan A. Orellana, Victoria Velarde, Juan C. Sáez, Gonzalo I. Gómez

**Affiliations:** 1Instituto de Ciencias Biomédicas, Facultad de Ciencias de la Salud, Universidad Autónoma de Chile, El Llano Subercaseaux #2801, Santiago 8910060, Chile; 2Departamento de Neurología, Escuela de Medicina y Centro Interdisciplinario de Neurociencias, Facultad de Medicina, Pontificia Universidad Católica de Chile, Santiago 8330024, Chile; 3Instituto de Neurociencias, Centro Interdisciplinario de Neurociencias de Valparaíso, Universidad de Valparaíso, Valparaíso 2381850, Chile; 4Departamento de Fisiología, Facultad de Ciencias Biológicas, Pontificia Universidad Católica de Chile, Alameda #340, Santiago 8331150, Chile

**Keywords:** connexin hemichannel, gap junction, oxidative stress, inflammatory receptors, Fasudil, Y-27632

## Abstract

Connexin 43 (Cx43) is expressed in kidney tissue where it forms hemichannels and gap junction channels. However, the possible functional relationship between these membrane channels and their role in damaged renal cells remains unknown. Here, analysis of ethidium uptake and thiobarbituric acid reactive species revealed that treatment with TNF-α plus IL-1β increases Cx43 hemichannel activity and oxidative stress in MES-13 cells (a cell line derived from mesangial cells), and in primary mesangial cells. The latter was also accompanied by a reduction in gap junctional communication, whereas Western blotting assays showed a progressive increase in phosphorylated MYPT (a target of RhoA/ROCK) and Cx43 upon TNF-α/IL-1β treatment. Additionally, inhibition of RhoA/ROCK strongly antagonized the TNF-α/IL-1β-induced activation of Cx43 hemichannels and reduction in gap junctional coupling. We propose that activation of Cx43 hemichannels and inhibition of cell–cell coupling during pro-inflammatory conditions could contribute to oxidative stress and damage of mesangial cells via the RhoA/ROCK pathway.

## 1. Introduction

Chronic kidney disease (CKD), defined as persistent alterations in kidney structure and/or function, has been recognized as a leading public health problem worldwide [1,2,3]. Regardless of the initiating insult or disease, the most common pathological manifestation of CKD is renal fibrosis [3]. The latter represents the unsuccessful wound-healing of kidney tissue after the chronic and sustained injury characterized by glomerulosclerosis, tubular atrophy, and interstitial fibrosis. Glomerulosclerosis occurs due to endothelial damage and dysfunction, the proliferation of smooth-muscle cells and mesangial cells (MCs), and the destruction of podocytes that line the glomerular basement membrane [1,2,3].

The glomerular MCs share most phenotypical similarities among renal cells with fibroblasts [4]. As one of the significant matrix-producing cells, MCs can secrete mesangial matrix components, such as type IV and type V collagens and fibronectin, which contribute to the excess extracellular matrix. Moreover, MCs also secrete several inflammatory cytokines, adhesion molecules, chemokines, and enzymes, all of which participate in the progression of renal fibrosis [4]. The primary pathological mechanism linking oxidative stress (OS), inflammation, and CKD progression includes an initial kidney injury caused by intra- and extracellular oxygen-derived radicals and the subsequent inflammatory response [5]. Indeed, superoxide and hydroxyl radicals readily interact with the molecular components of nephrons [5]. Tumor necrosis factor-α (TNF-α) initiates the proinflammatory response by binding to its receptors, TNFR1 and TNFR2, expressed on tubular (and other) cell surfaces, triggering pathways that activate nuclear factor κB (NFκB) and downstream signaling [6]. Similar effects have been attributed to IL-1β in human and experimental kidney diseases [7]. Recently, in an experimental model of type II diabetes, the IL-1β receptor antagonist, anakinra, presented protective effects, unveiling the participation of this cytokine in renal inflammation [7]. During kidney injury, MCs are the principal target of angiotensin II (AngII) and participate in inflammatory reactions and OS, which lead to CKD involving an increase in intraglomerular pressure [8,9]. Consequently, within the glomerulus, the activation of MCs, epithelial cells, and podocytes elicits the release of vasoactive and proinflammatory agents that increase cell damage and promote fibrosis. This not only causes a reduction in renal blood flow, but also leads to deficits in permeability during filtration [10]. Several studies have shown that the inflammatory response requires reactive oxygen species (ROS) to link AngII with downstream production of TNF-α, IL-1β, and IL-6 [2,9].

The renin–angiotensin system (RAS) is a vital endocrine control system in mammals, with AngII being the primary active component [11]. AngII binds to several receptors, including the two specific and ubiquitous cytoplasmic angiotensin membrane G-protein-coupled receptors denominated type I (AT1) and type II (AT2), to carry out many biological functions [12]. RAS is one of the crucial regulatory systems controlling blood pressure and fluid balance and comprises two principal axes: the conventional and non-conventional [13,14]. The principal promoter of the RAS conventional axis is AngII, which is released by local and systemic RAS activation further to stimulate its receptor, AngII type1 receptor (AT1R). The latter results in renal vasoconstriction, inflammation and tubular damage accompanied by tubulointerstitial ischemia following a glomerular lesion, and hyperfunction of the remaining tubules, promoting recruitment of leukocytes, cytotoxicity, and fibrogenesis, with consequences for the progression of several kidney diseases [14]. These include nephropathies, renal artery stenosis, and acute kidney injury [13]. On the other hand, the non-conventional or protective axis of RAS induces vasodilation in various tissues such as the heart, kidney, lung, and blood vessels [13,14].

Coupled to a G-protein, AT1 and AT2 receptors mediate the actions of Ang II. The AT1 receptor activates small G proteins, including Ras, Rac1, RhoA, and the Rho kinase system (ROCK) [15], while the AT2 receptor inhibits RhoA [16]. This major group of small GTPases, the Rho GTPases (average molecular weight 20–40 kDa), regulate cell junction, cytoskeleton, and cell migration [11]. RhoA is the most recognized member of the Rho GTPase family. ROCK, which exists in two isoforms, ROCK1 and ROCK2, is a downstream effector of RhoA [17]. ROCK is critical in controlling migration, proliferation, cell apoptosis/survival, gene transcription, and differentiation [17]. The RhoA/ROCK signaling pathway plays an essential role in cell differentiation, migration, growth, and development [18]. The alteration of this signaling pathway can lead to various diseases, including renal interstitial fibrosis caused by unilateral ureteral obstruction [18]. Regarding a therapeutic approach, Fasudil and Y-27632, two non-selective ROCK1/2 inhibitors, have been used to evaluate the role of ROCK in several animal disease models [19]. For example, in a rat heart ischemia-reperfusion model, Fasudil reduced the infarct size by attenuating endoplasmic reticulum stress and modulating the activity of sarco/endoplasmic reticulum Ca^2+^-ATPase [19]. However, the role of the RhoA/ROCK pathway in the regulation of adhesion and inflammation in the glomerulus remains unsubstantiated [17].

Inflammation encompasses a myriad of complex mechanisms and cellular processes, making dysfunctional communication among cells a critical factor. In vertebrates, part of the intercellular communication takes place through gap junctions (GJs) and hemichannels (HCs) [9]. GJs are clusters of intercellular channels, each resulting from the docking of two HCs or connexons at cell–cell interfaces. HCs are formed by the oligomerization of six protein subunits called connexins (Cxs) around a central pore. Thus, GJs mediate communication via the cytoplasm of adjacent cells, whereas undocked HCs allow the diffusion of substances between the cytoplasm and extracellular space [9]. Both GJs and HCs are permeable to ions and small molecules, permitting the coordination and regulation of different biological processes [9]. There are 20 and 21 Cx isoforms in humans and rats, respectively [20], with Cx43 being the main isoform in vascular endothelial cells playing a fundamental role in regulating vascular diseases such as atherosclerosis, hypotension, and bradycardia [21]. The existence of intercellular GJ channels in the kidney was demonstrated about three decades ago. Several studies have provided evidence of the expression of nine Cx isoforms in the kidney, including Cx26, Cx30, Cx30.3, Cx32, Cx37, Cx40, Cx43, Cx45, and Cx46 [22,23]. Therefore, these Cx-based channels could play an essential role in several renal functions such as maintenance of acid–base homeostasis, reabsorption and secretion of metabolites, glomerular filtration, and regulation of blood pressure through water reabsorption and renin secretion.

In vitro data from renal epithelial cells exposed to high glucose provided a functional explanation for Cx43 upregulation under stressful conditions [21]. This evidence suggests that Cx43 GJs mediate the intercellular transfer of deleterious Ca^2+^ signals for proper cell function [21]. On the other hand, it was found that MES-13 cells, a line derived from MCs, stimulated with AngII, developed oxidative stress, secreted proinflammatory cytokines (IL-1β and TNF-α), and showed a progressive increase in the activity of HCs. In addition, Western blotting analysis showed that phosphorylated MYPT (a substrate of RhoA/ROCK pathway) and Cx43 increased progressively upon AngII treatment, suggesting a possible relationship between the RhoA/ROCK pathway and Cx43 [9]. However, the precise role of Cx43 in the progression of renal disease remains unclear. Here, we found that TNF-α plus IL-1β increases OS and Cx43 HC activity in MCs, and these effects were prevented by Fasudil or Y-27632, both inhibitors of the RhoA/ROCK pathway.

## 2. Results

### 2.1. TNF-α/IL-1β Induced Activation of Cx43 HCs in Mesangial Cells Depends on RhoA/ROCK Pathway

We previously demonstrated that activation of Cx43 HCs evoked by AngII occurs at the same time as the high production of IL-1β and TNF-α in MES-13 cells [9]. However, whether both cytokines directly increase the activity of Cx43 HCs in these cells remained to be elucidated. For that purpose, the functional state of HCs was investigated by recording the rate of Etd^+^ uptake. We found that under control conditions, MES-13 and primary MCs cells display a low Etd^+^ uptake. In contrast, the treatment for 72 h with TNF-α/IL-1β (10 ng/mL) induced an increase in Etd^+^ uptake compared to control values (from 3.2 ± 1.0 to 16.8 ± 1.1 AU/min, in MES-13; and from 1.0 ± 0.2 to 1.8 ± 0.3-fold change of control in primary MCs, respectively) (Appendix A; Figure 1A). No change in Etd^+^ uptake was observed in cells that were stimulated for less than 72 h or when TNF-α or IL-1β was added alone (Appendix A).

Since Cx43 HCs represent one of the most prevalent routes for dye influx in MES-13 cells [9]. The potential contribution of these channels in the TNF-α/IL-1β-induced Etd^+^ uptake was examined. Accordingly, primary MCs cultures were pre-incubated for 15 min before and throughout Etd^+^ uptake recordings with various pharmacological agents. We found that 200 µM La^3+^, a general blocker of Cx HCs [24], induced a reduction in the Etd^+^ uptake evoked by TNF-α/IL-1β (1.1 ± 0.2-fold change of control). Interestingly, 100 µM Gap19 and TAT-L2, two Cx43 HC inhibitory mimetic peptides with sequences equivalent to the intracellular loop domain of Cx43 [25], completely blunted the TNF-α/IL-1β-induced Etd^+^ uptake in primary MCs (0.9 ± 0.1 and 0.9 ± 0.1-fold change of control, respectively) (Figure 1A,B). Overall, these results reveal that TNF-α/IL-1β enhances the function of Cx43 HCs in MCs.

To assess whether TNF-α/IL-1β-induced opening of Cx43 HCs in MCs is mediated by a RhoA/ROCK-dependent pathway, we evaluated the effect of Fasudil—selective inhibitor of ROCK [11,15,26]. Similarly, pretreatment of MCs with Fasudil (15 µM) reduced the TNF-α/IL-1β-induced activity of Cx43 HCs (1.0 ± 0.1-fold change of control) (Figure 1). The latter suggests that the increase in Cx43 HCs caused by TNF-α/IL-1β relies on ROCK activation in MCs.

### 2.2. TNF-α/IL-1β Reduces Intercellular Communication Mediated by GJs in Mesangial Cells

Multiple lines of research have described that under proinflammatory conditions, the increased activity of HCs occurs in parallel with a decrease in gap junctional communication [27,28,29]. With this in mind, we decided to explore whether TNF-α/IL-1β could affect the functional state of GJs in MES-13 cells by measuring the intercellular diffusion of microinjected Etd^+^ on single cells grown in clusters or monolayers [30].

Etd^+^ intercellular coupling experiments revealed that under control conditions, almost 100% of MCs were coupled, in general with ~6 neighboring cells (Figure 2A–C). It is worth noting that 72 h of treatment with TNF-α/IL-1β caused a prominent reduction in the incidence of coupling (44 ± 6%) (Figure 2C). In addition, the number of coupled cells (coupling index) decreased from 6.0 ± 0.2 cells to 2.0 ± 0.2 cells after treatment with TNF-α/IL-1β (Figure 2B). Altogether, these findings indicate that TNF-α/IL-1β impacts the activity of HCs and GJ channels in an opposite manner. Surprisingly, pretreatment with 15 µM Fasudil strongly prevented the inhibitory effect of TNF-α/IL-1β on gap junctional communication (incidence of coupling: 86.0 ± 3.8%; coupling index: 4.0 ± 0.4 cells) (Figure 2A–C). No changes in cell–cell coupling was seen in cells treated with Fasudil alone (Figure 2A).

Given that the increased HC activity induced by inflammatory conditions occurs along with the decrease in dye coupling in many cell types [27], we also investigated whether the functional state of GJs was affected by TNF-α/IL-1β in primary MCs. Control MCs exhibited high LY intercellular diffusion (Figure 3A). Nonetheless, 72 h after treatment with TNF-α/IL-1β, intercellular dye transfer decreased compared with MCs under control conditions (from 1.0 ± 0.0 to 0.4 ± 0.1-fold change of control) (Figure 3B). Similarly, treatment with Fasudil completely prevented the MC uncoupling triggered by TNF-α/IL-1β (1.0 ± 0.1-fold change of control) (Figure 3B). These findings reveal that the same metabolic pathway that increases the activity of HCs seems to be involved in the reduction of cell–cell communication mediated by GJs between MCs.

### 2.3. TNF-α/IL-1β Promotes Phosphorylation of MYPT and Increases the Amount of Cx43 in Mesangial Cells

Given that TNF-α/IL-1β activates RhoA and Rho kinase (ROCK) [31] and alters Cx43 levels in different cell types [32,33,34], we decided to evaluate the activity of RhoA/ROCK and Cx43 protein levels. Accordingly, we first measured the amount of phosphorylated MYPT—a downstream effector of the RhoA/ROCK pathway—and the relative amount of Cx43 in MES-13 cells after treatment with TNF-α/IL-1β.

A clear increase in the relative amount of Cx43 was found in MES-13 cells after 72 h of treatment with TNF-α/IL-1β compared to control conditions. This response was partially inhibited by pretreatment with either Fasudil (0.7 ± 0.1 AU) or Y-27632 (0.6 ± 0.0 AU) (Figure 4A,B). A similar effect was observed on the distribution of Cx43 in confluent primary MCs, the latter measured by immunofluorescence analysis (Figure 4C). In addition, 72 h of treatment with TNF-α/IL-1β also increased the phosphorylation of MYPT (1.0 ± 0.1 AU) compared to control conditions (Ctrl 0.3 ± 0.0 AU), a response partially suppressed by Fasudil or Y-27632 (0.6 ± 0.1 AU or 0.5 ± 0.1 AU) (Figure 4D,E). Of note, neither Fasudil nor Y-27632 affected Cx43 levels or phosphorylation of MYPT in control cells (Figure 4D,E). Therefore, these data indicate that increases in phosphorylated MYPT and Cx43 levels evoked by TNF-α/IL-1β could be partly explained by the activation of RhoA/ROCK- pathway.

### 2.4. Inhibition of RhoA/ROCK Prevents Increases in Lipid Peroxidation Responses Induced by TNF-α and IL-1β in Mesangial Cells

TNF-α/IL-1β induces Ca^2+^ influx from the extracellular space and Ca^2+^ release from intracellular stores [35], leading to ROS generation and cell damage in several kidney diseases [36,37]. In addition, HCs regulate the cytosolic Ca^2+^ concentration, as they are permeable to this divalent cation or facilitate its intracellular increase by releasing ATP that activates purinergic receptors [38,39]. In this scenario, we evaluated whether TNF-α/IL-1β could cause OS in MCs. We observed that the extracellular amount of TBARS increased (4.6 ± 0.2 µmol/L) in primary MCs treated with TNF-α/IL-1β compared to control conditions (1.7 ± 0.0 µmol/L) (Figure 5). Moreover, when Fasudil was added to cells treated with TNF-α/IL-1β, the extracellular amount of TBARS was significantly lower (TNF-α/IL-1β+ Fasudil; 1.6 ± 0.1 µmol/L), and similar to what was found in control cell culture treated with Fasudil (1.5 ± 0.0 µmol/L) (Figure 5). These data suggest that a RhoA/ROCK-dependent pathway increases OS in MCs.

### 2.5. Inhibition of RhoA/ROCK Prevents Apoptosis and Cell Viability Induced by TNF-α and IL-1β in Primary Mesangial Cells

Given that TNF-α/IL-1β activates RhoA/Rho kinase (ROCK) [31] and Cx43 HCs-mediated cellular damage in MES-13 cells [9], the potential contribution of both in TNF-α/IL-1β-induced apoptosis and cellular viability was examined. MCs treated for 72 h with TNF-α/IL-1β presented a significant reduction in cell viability (72.0 ± 0.0%) compared to control conditions (100.0 ± 0.0%). Moreover, when Fasudil was added to cells treated with TNF-α/IL-1β, the viability remained high (TNF-α/IL-1β plus Fasudil; 90.0 ± 0.2%), and comparable to what was found in control cell culture treated with Fasudil (88.0 ± 0.1%) (Figure 6A). These data indicate that a RhoA/ROCK-dependent pathway increases OS in primary MCs, which negatively affects cell viability. Furthermore, we investigated whether the enhanced activity of HCs induced by TNF-α/IL-1β in MCs could promote apoptosis. Under control conditions, MCs were not labeled with Tunel staining (DNAase was used as a positive control; Appendix A). However, after 72 h treatment with TNF-α/IL-1β, a prominent increase in apoptosis was observed—a response strongly prevented by Fasudil added 24 h before the end of the experiment (Figure 6B,C).

## 3. Discussion

The latest epidemiological study published in 2020 places CKD as the 12th leading cause of death worldwide, with a prevalence that increases each year [40], making this disease one of the leading public health problems today. It is essential, therefore, to further understanding CKD’s pathological mechanisms and to develop new therapeutic approaches [40]. In this work, we demonstrated that TNF-α/IL-1β increases Cx43 HC activity and simultaneously reduces gap junctional communication in MCs. Noticeably, the above response was strongly prevented by inhibiting the RhoA/ROCK pathway, indicating its crucial role in the TNF-α/IL-1β-mediated modulation of Cx-based channels in MCs. Of note, blockade of RhoA/ROCK pathway also reduced the TNF-α/IL-1β-induced production of OS. Based on this, we propose that RhoA/ROCK signaling contributes to the TNF-α/IL-1β induced modulation of HCs and GJs with significant, potentially negative consequences for the function and survival of MCs (Figure 7).

MCs are pivotal for the normal functioning of the glomerulus [41]. These cells regulate intraglomerular capillary flow and the ultrafiltration surface due to their contractile properties and ability to respond to different substances. Among them are vasoactive molecules such as prostaglandin, adenosine, vasopressin, norepinephrine, AngII [41,42], and inflammatory mediators including IL-1β, TNF-α, and IFN-γ [42]. In addition, MCs provide the structural support for the glomerular capillary network, and their crosstalk with other glomerular cell types, such as podocytes and endothelial cells, is fundamental for the proper function of the kidney [43]. Moreover, MCs play a role in the innate renal immune response as phagocytic cells by eliminating macromolecules, cells, and apoptotic bodies present in the mesangium [42]. Finally, MCs generate and control the turnover of the mesangial matrix in response to environmental cues [44,45]. Therefore, due to the multiple roles of MCs, they are considered a critical element in the origin and progression of various kidney diseases.

The pathophysiological mechanisms of kidney diseases are associated with factors that predispose to redox imbalance and the generation of inflammatory mediators, including ROS, TNF-α and IL-1β [5,46,47]. Inflammation is a well-known condition that reduces cell–cell coupling but increases HC activity [32,33,34], and several studies have found that GJs and HCs are regulated in opposite ways; for Cx43-based channels, it has been suggested that interaction of the C terminus with the cytoplasmic loop distinctly influences the function of GJs and HCs [48]. Consistent with this, the treatment with TNF-α/IL-1β for 72 h increased the Etd^+^ uptake rate in MCs, a response being prevented by the blockade of Cx43 HCs with Gap19 and a TAT-L2. Likewise, TNF-α/IL-1β also reduced GJ-mediated cell–cell coupling and increased the amount of Cx43 protein in MCs. Previous studies have demonstrated that models of hypertensive CKD and inflammation CKD increase the expression of Cx43 in the early stage of CKD in the glomerulus [49], whereas increased HC activity occurs during heart attack and failure [50,51], neurodegenerative diseases [52], and liver fibrosis [53]. Several studies have shown that TNF-α/IL-1β canonically activates NFκB, a critical transcriptional factor that underpins the inflammatory response and has substantial consequences for redox balance as well [54]. Interestingly, the promoter of Cx43 contains a positive regulatory binding site for NFκB [55,56]. On the other hand, NFκB also induces the expression of iNOS and Cx43 via activation of PKA in MCs [57]. The latter would partially explain the increase in Cx43 protein observed in diseases with increased proinflammatory factors. Furthermore, nitric oxide (NO) increases the opening probability of Cx43 HCs by S-nitrosylation of cysteine at position 271, without altering their phosphorylation state [58,59]. Recently and similarly to what occurs in cortical astrocytes treated with TNF-α and IL-1β [27] metabolic inhibitors, a reduced intercellular communication mediated by GJs and increased membrane permeability through HCs formed by Cx43 have been found in cultures of proximal tubule cells [60,61]. Therefore, it is possible to speculate that Cx43 HCs can be considered a new mediator of renal disease involved in central processes of inflammation and fibrosis. Their inhibition, even after the initiation of the disease, attenuates renal damage and preserves renal function in animal models of vascular, tubular, and glomerular CKD [21]. Thus, inhibition of Cx43 HCs represents a promising beneficial effect to reduce inflammation and fibrosis, maintaining tissue integrity [62,63] (Figure 7).

RhoA/ROCK-dependent signaling pathways are critically involved in pathological conditions, including pulmonary hypertension [64], heart attack [65], stroke [66], Alzheimer’s disease [67], glaucoma [68], diabetes and hypertensive nephropathy [69,70]. In addition, RhoA/ROCK is activated in MCs after stimulation with AngII [14]. Consistent with the above, we found an increase in levels of MYPT phosphorylated after treatment with TNF-α/IL-1β. These results are consistent with studies carried out in endothelial cells, where stimulation with TNF-α induces the activation of the RhoA/ROCK pathway by modulating the cytoskeleton and JNK-dependent secretion of IL-6 [71].

Intercellular communication in the kidney occurs directly via the cytoplasm of adjacent cells connected through GJs, and by paracrine signals released via large-pore channels, such as HCs, pannexons and P2X_7_Rs [72]. The release of ATP via HCs has multiple functions in the kidney, including regulating renal blood flow, glomerular filtration rate, and renal tubular transport [73]. However, the HC-mediated release of ATP and further activation of P2 × _7_Rs have been linked to inflammation and fibrosis [72,74]. Multiple studies argue that proinflammatory cytokines may contribute to a chronic activation of endothelial cells, and thereby, a long-term production of key “danger” signals, such as ATP [20,75,76,77]. The intensity of this response might impact the outcome of the inflammation. In that regard, it has been demonstrated that opening of Cx43 hemichannels could lead to preconditioning [78] as well as to cell death [20,28]. Interestingly, our results show that inhibition of the RhoA/ROCK pathway prevents: (i) the loss of GJ-mediated coupling, (ii) the increase in Cx43 HC activity and (iii) the increase in Cx43 protein levels observed after TNF-α/IL-1β treatment. Therefore, the latter signaling is crucial for regulating the expression and function of Cx43-based channels in MCs during pro-inflammatory conditions. These results are consistent with previous evidence showing that the increased HC activity evoked by stimulation with AngII depends on activating RhoA/ROCK signaling in MES-13 cells [9], and recently it has also been reported that RhoA/ROCK activation enhances Cx43 HC function [79] (Figure 7). Relevantly, studies carried out in corneal epithelial cells in inflammatory conditions have shown that RhoA/ROCK signaling participates in the loss of cell–cell communication mediated by Cx43 GJ channels [80]. A similar effect has been observed in fibroblasts, where tissue stretching causes the opening of Cx43 HCs and the release of ATP through a mechanism mediated by RhoA/ROCK signaling [81]. Contrarily, thrombin-induced activation of RhoA GTPase controls extracellular purinergic signaling in endothelial cells by inhibiting Cx43 HCs [82]. Despite the apparent relationship between RhoA/ROCK signaling and Cx43, the use of Fasudil and Y-27632 did not reach the Etd^+^ uptake values achieved by blocking of Cx43 HCs with Gap19 and TAT-L2. The latter could be due to the existence of additional mechanisms that would increase the activity of HCs in a manner independent of RhoA/ROCK, such as the production of NO by iNOS [58].

The RhoA/ROCK-dependent signaling pathway has been extensively investigated in hypertensive pathology, where it plays a crucial role in regulating blood pressure and peripheral resistance [70]. In fact, treatment with Fasudil and Y-27632 has been suggested as an anti-hypertensive treatment given their hypotensive effect in DOCA-salt rat models [83] and spontaneously hypertensive rats [84]. Nevertheless, the NFκB pathway can also be activated by proteins of the Rho family: RhoA, Rac1 and Cdc42, which participate downstream of IL-1β and TNF-α [85,86]. This work provides information to understand the possible molecular mechanisms underlying the dysfunction and damage of the kidney during pathological conditions and chronic diseases.

## 4. Materials and Methods

### 4.1. Reagents

TNF-α and IL-1β were obtained from Alomone (Jerusalem, Israel); Fasudil, Y-27632, ethidium (Etd^+^) bromide and lanthanum (La^3+^) chloride were obtained from Sigma-Aldrich (St. Louis, MO, USA). TBARS assay kit was obtained from Cayman Chemicals (Ann Arbor, MI, USA), and CellTiter96^®^ Non-Radioactive Cell Proliferation Assay was obtained from Promega (Madison, WI, USA). The mimetic peptides gap19 (KQIEIKKFK, intracellular loop domain of Cx43) and TAT-L2 (YGRKKRRQRRRDGANVDMHLKQIEIKKFKYGIEEHGK, second intracellular loop domain of Cx43) were obtained from GenScript (Piscataway Township, NJ, USA) [25]. The monoclonal anti-α-tubulin antibody was purchased from Sigma-Aldrich (St. Louis, MO, USA). The polyclonal anti-phosphorylated-MYPT1 (Thr696) antibody was obtained from Merck Millipore (Darmstadt, Germany). The monoclonal anti-MYPT1 antibody was purchased from BD Transduction Laboratories (San José, CA, USA). The monoclonal anti-unphosphorylated Cx43 antibody was obtained from Invitrogen (Carlsbad, CA, USA). Anti-mouse and anti-rabbit secondary antibodies conjugated to horseradish peroxidase were from Santa Cruz Biotechnology Inc. (Santa Cruz, CA, USA).

### 4.2. Experimental Animals and Isolation of Primary Glomerular MCs

C57BL/6 (U. Chile) mice of 12–16 weeks of age were housed in cages in a temperature-controlled (24 °C) and humidity-controlled vivarium under a 12 h light/dark cycle (lights on 8:00 a.m.), with ad libitum access to food and water. All procedures were in accordance with institutional and international standards for the humane care and use of laboratory animals (Animal Welfare Assurance Publication A5427-01, Office for Protection from Research Risks, Division of Animal Welfare, NIH (National Institutes of Health), Bethesda, MD, USA). The Bioethical and Biosafety Committee of the Faculty of Biomedical Sciences at Universidad Autónoma de Chile (BE07-20; 26 October 2020) approved the described experimental procedures.

Primary MCs were isolated from mouse glomeruli treated with collagenase [87,88,89,90]. In brief, the kidney fragments were minced with a razor blade, and a 100 μm nylon sieve was used to collect the glomeruli from the cortex homogenates of mice kidneys in aseptic conditions. This glomeruli-enriched fraction was collected from underneath the sieve with HBSS. The diluted suspension was poured onto a second 70 μm filter and washed with the same solution. The glomeruli and other fragments retained on the filter were transferred into a sterile tube. The glomerular suspension was incubated for digestion with sterile type IV collagenase in DMEM for 1 h at 37 °C in an incubator. Then, it was triturated through a 21-gauge needle. Glomerular remnants were washed, and mesangial and endothelial cells were plated onto a six-well plate in complete medium and incubated at 37 °C in a humidified 5% CO^2^ incubator [87,88,89,90].

### 4.3. Cell Cultures

The cell line MES-13, derived from mesangial cells (CRL-1927 from ATCC, Manassas, VA, USA) or isolated mesangial cells, were cultured and treated with TNF-α plus IL-1β (10 ng/mL of each one) for different time periods (0, 24, 48 and 72 h) in a 2:1 mixture of DMEM and F-12 tissue culture media or MEM supplemented with 100 U/mL penicillin and 100 μg/mL streptomycin. Cells were kept at 37 °C in 5% CO_2_/95% air, at nearly 100% relative humidity. Fasudil (15 μM) or Y-27632 (15 μM) were added 24 h before the end of a three-day experiment to cell cultures treated with TNF-α and IL-1β at time zero.

### 4.4. Dye Uptake and Time-Lapse Fluorescence Imaging

The activity of Cx HCs was evaluated by using the dye uptake method, as previously described [91]. In brief, cells at 70% confluence were plated onto glass coverslips and bathed with Locke’s saline solution (in mM: 154 NaCl, 5.4 KCl, 2.3 CaCl_2_, 1.5 MgCl_2_, 5 HEPES, 5 glucose, and pH 7.4) containing 5 μM ethidium bromide (Etd^+^), a molecule that crosses the plasma membrane through large-pore channels, including HCs [91]. Since Etd^+^ fluoresces upon its intercalation between nucleotides of the DNA, time-lapse recordings of fluorescent images were measured (at regions of interest in different cells) every 30 s for 13 min using a Nikon Eclipse Ti inverted microscope (Tokyo, Japan) and NIS-Elements software. The basal fluorescence signal was recorded in cells only in Locke’s saline solution that contained divalent cations. The fluorescence intensity recorded from 25 regions of interest (representing 25 cells per coverslip) was defined as the subtraction (F–F0) between the fluorescence (F) from the respective cell (25 cells per field) and the background fluorescence (F0) measured where no labeled cells were detected. The mean slope of the relationship F-F0 over a given time interval (ΔF/ΔT; F0 remained constant along the recording time) represents the Etd^+^ uptake rate. To determine changes in slope measurements, regression lines were fitted to points before and after the various experimental conditions using Excel software, and mean values of slopes were compared using GraphPad Prism software and expressed as AU/min. At least four replicates (four sister coverslips) were measured in each independent experiment [20].

### 4.5. Dye Coupling

MES-13 cells seeded on glass coverslips (n^o^1) were bathed with Locke’s saline solution, and then observed using an inverted microscope equipped with xenon arc lamp and a Nikon B filter (excitation wavelength: 450–490 nm, emission wavelength: above 520 nm). Etd^+^ (25 mM) was microinjected through a glass microelectrode into one cell. Dye transfer to neighboring cells was evaluated 2 min after injection. We routinely performed all dye coupling experiments in the presence of La^3+^ (150 µM) to prevent Etd^+^ leakage through HCs that would reduce the intercellular diffusion among coupled cells [92]. The incidence of dye coupling was scored as the percentage of injections that resulted in dye transfer from the injected cell to more than one neighboring cell. The coupling index was calculated as the average number of cells to which the dye had spread, divided by the number of positive cases. Four experiments were performed for every treatment, and dye coupling was tested by microinjecting a minimum of 10 cells per experiment.

### 4.6. Scrape Loading/Dye Diffusion Technique

GJ permeability was evaluated at room temperature (RT) using the scrape loading/dye transfer (SL/DT) technique [93,94]. Briefly, MCs cultures were washed for 10 min in HEPES-buffered salt solution containing the following (in mM): 140 NaCl, 5.5 KCl, 1.8 CaCl_2_, 1 MgCl_2_, 5 glucose, 10 HEPES, pH 7.4, followed by washing in a Ca^2+^-free HEPES solution for 1 min. Then, a razor blade cut was made in the monolayer in a HEPES-buffered salt solution with normal Ca^2+^ concentration containing the fluorescent dye Lucifer yellow (LY). After 1 min, LY (100 μM) was washed out several times with HEPES buffered salt solution. At 8 min after scraping, fluorescent images were captured using a Zeiss Axio Observer D.1 Inverted Microscope with a Solid-State Colibri 7 LED illuminator and with a 10× objective. Changes were monitored using an AxioCam MRm monochrome digital camera R3.0 (Carl Zeiss AG, Zeiss, Oberkochen, Germany), and Software ZEN Pro (Zen 2.3 [blue edition], Carl Zeiss AG, Oberkochen, Germany) for image acquisition and analysis. For each trial, data were quantified by measuring fluorescence areas in three representative fields. Quantification of changes in GJ communication induced by different treatments was performed by measuring the fluorescence area, expressed as AU [93,94].

### 4.7. Immunofluorescence

MCs grown on coverslips were fixed at RT with 2% paraformaldehyde for 30 min and then washed three times with PBS. They were incubated three times for 5 min in 0.1 M PBS glycine, and then in 0.1% PBS-Triton X-100 containing 10% NGS for 30 min. Cells were incubated with anti-PDGFRB monoclonal antibody (Invitrogen, 1:100) and anti-Cx43 polyclonal antibody (#C6219 SIGMA, 1:100) diluted in 0.1% PBS-Triton X-100 with 2% NGS at 4 °C overnight. After five rinses in 0.1% PBS-Triton X-100, cells were incubated with goat anti-mouse IgG Alexa Fluor 488 (1:1000) or goat anti-rabbit IgG Alexa Fluor 555 (1:1000) at RT for 60 min. After several rinses, coverslips were mounted in Paramount-DAPI fluorescent mounting medium and examined with high-resolution fluorescence microscopy (Leica, Wetzlar, Germany) with a 63X objective.

### 4.8. Western Blot Assays

Cell cultures were placed on ice, washed twice with ice-cold PBS (pH 7.4), and harvested by scraping in 80 μL of a solution containing a protease and phosphatase inhibitor cocktail (Thermo Scientific, Pierce, Rockford, IL, USA; cat # 78430). Lysates were centrifuged (25,200× *g*, Eppendorf Centrifuge 5415C, Hamburg, Germany), and supernatants were collected for Western blot analysis. Protein concentration was determined using Lowry’s method [95]. Samples of homogenized cell cultures (50 μg of proteins) under different conditions were resolved by electrophoresis in 10% SDS-polyacrylamide gel, and in one lane pre-stained molecular weight markers were resolved. Proteins were transferred to a PVDF membrane (pore size: 0.45 μm), which was blocked at RT with Tris pH 7.4, 5% skim milk (*w/v*) and 1% BSA (*w/v*). Then, the PVDF membrane was incubated overnight at 4 °C with anti-Cx43 (1:1000), anti-p-MYPT1 (#ABS45 Merck Millipore, 1:500), or anti-MYPT1 (#612164 BD Transduction Laboratories, 1:1000) antibody, followed by incubation with rabbit or mouse secondary antibody conjugated to peroxidase (1:2000 both) for 1 h at RT. Then, the PVDF membrane was stripped and reblotted with the anti-α-tubulin antibody (1:5000) used as loading control, following the same procedure described above. After repeated rinses, immunoreactive proteins were detected by using ECL reagents (Pierce Biotechnology, Rockford, IL, USA) according to the manufacturer’s instructions. The bands detected were digitized and subjected to densitometry analysis using the software ImageJ (Version 1.50i, NIH, Washington, DC, USA).

### 4.9. Thiobarbituric Acid Reactive Substances (TBARS) Measurement

The amount of TBARS was estimated using the method described by Ramanathan and collaborators [96] with slight modifications. Culture medium was mixed with SDS (8% *w/v*), thiobarbituric acid (0.8% TBA *w/v*), and acetic acid (20% *v/v*), and heated for 60 min at 90 °C. The material that precipitated was removed by centrifugation, and the absorbance of the supernatant was evaluated at 532 nm. The amount of TBARS was calculated using a calibration curve obtained with malondialdehyde (MDA) as standard. MDA was obtained from Merck (Darmstadt, Germany).

### 4.10. TUNEL Assay

Apoptosis was assessed using Click-iT TUNEL Alexa Fluor Imaging Assay as recommended by the supplier (C-10246; Invitrogen). Briefly, MCs were cultured in a 5% CO_2_ incubator at 37 °C. Then, cells were fixed with 4% paraformaldehyde in PBS for 1 h at RT. Fixed cells were treated with permeabilization solution (0.1% Triton X-100) for 2 min at RT and then incubated with 50 μL of TUNEL reaction buffer for 1 h in a 37 °C humidified atmosphere in the dark. After incubation, cells were treated with 50 μL of converter-POD (anti-fluorescein antibody, Fab fragment from sheep, conjugated with horse-radish peroxidase) in a 37 °C humidified chamber for 30 min and then treated with 50 μL DAB (3,3′-diaminobenzidine) substrate for 10 min at RT. Percentage of apoptotic cells was estimated by counting TUNEL-positive red cells divided by the number of ≥15 cells for field. For each trial, data were quantified by measuring fluorescence in five representative fields using high-resolution fluorescence microscopy (Leica, Wetzlar, Germany).

### 4.11. Cell Viability

The number of viable cells was quantified using an MTT/PMS reagent-based Cell Titer 96 Aqueous Non-Radioactive Cell Proliferation Assay Kit (Promega) according to the manufacturer’s instructions [97].

### 4.12. Statistical Analysis

For each data group, results were expressed as mean standard error (SEM); n refers to the number of independent experiments. For statistical analysis, each treatment was compared with its corresponding control, and significance was determined using a one-way ANOVA followed, in case of significance, by a Tukey post hoc test. Analyses were performed with the GraphPad Prism 9 software for Windows (1992–2020, GraphPad Software, La Jolla, CA, USA).

## Figures and Tables

**Figure 1 ijms-23-10097-f001:**
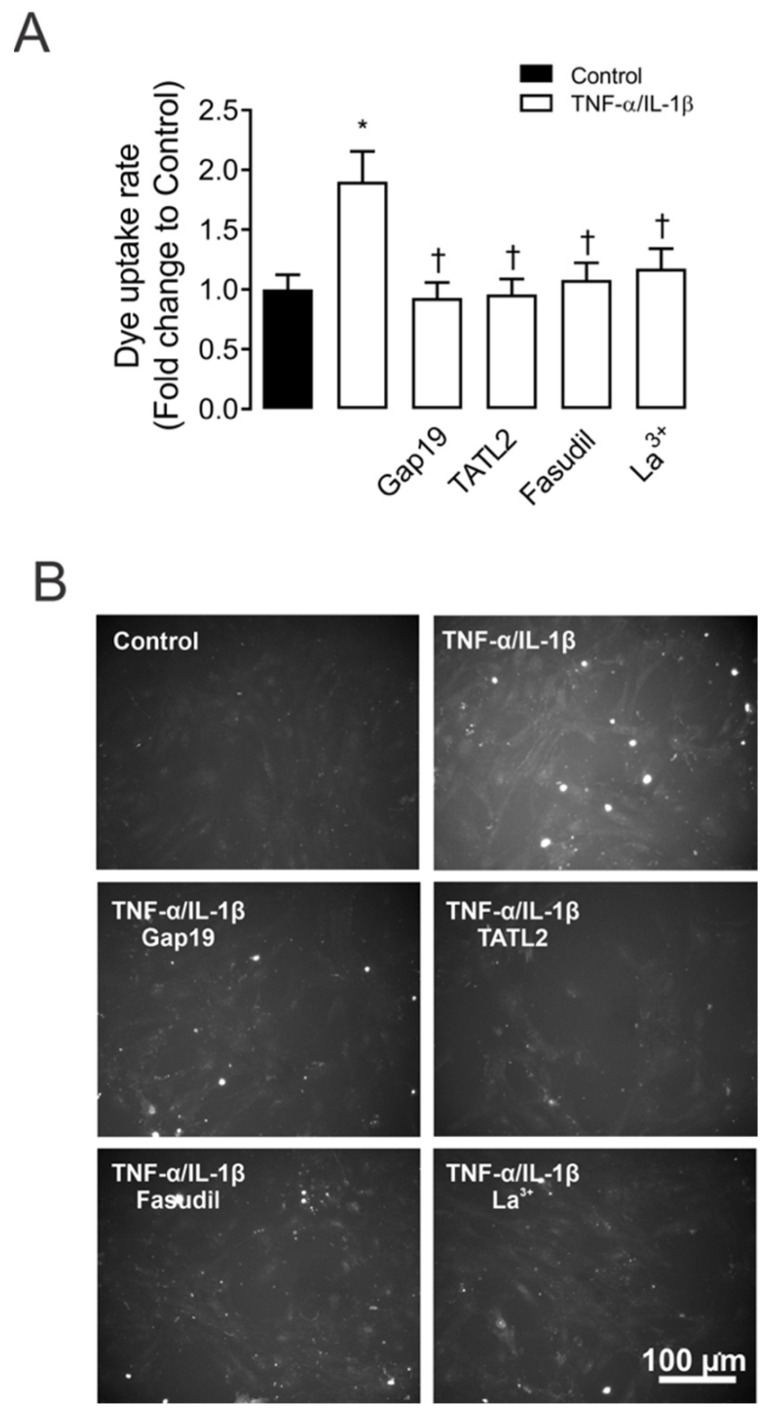
ROCK and Cx43 HC blockers reduce the TNF-α/IL-1β-induced Etd^+^ uptake in mesangial cells. (**A**) Etd^+^ uptake rate in primary MCs under control conditions (Ctrl, black bar), exposed to TNF-α/IL-1β (10 ng/mL each, white bars) for 72 h alone or with ROCK inhibitor: Fasudil (15 μM); and HCs blockers: Lanthanum ion (La^3+^, 200 μM) and selective Cx43 HC blockers: mimetic peptides Gap19 (100 µM) and TAT-L2 (100 µM). (**B**) Fluorescence images show primary MCs exposed to ethidium (5 μM Etd^+^) for 13 min under TNF-α/IL-1β stimulation with or without different blockers. Scale bar = 50 μm. Each bar represents the mean value ± SEM of five independent experiments with four replicates each. Statistical significance * *p* < 0.05 vs. Ctrl; † *p* < 0.05 vs. TNF-α/IL-1β. Fasudil was added during the last 24 h of treatment. Gap19, TAT-L2 and La^3+^ were added acutely (added 15 min before each Etd^+^ uptake recording).

**Figure 2 ijms-23-10097-f002:**
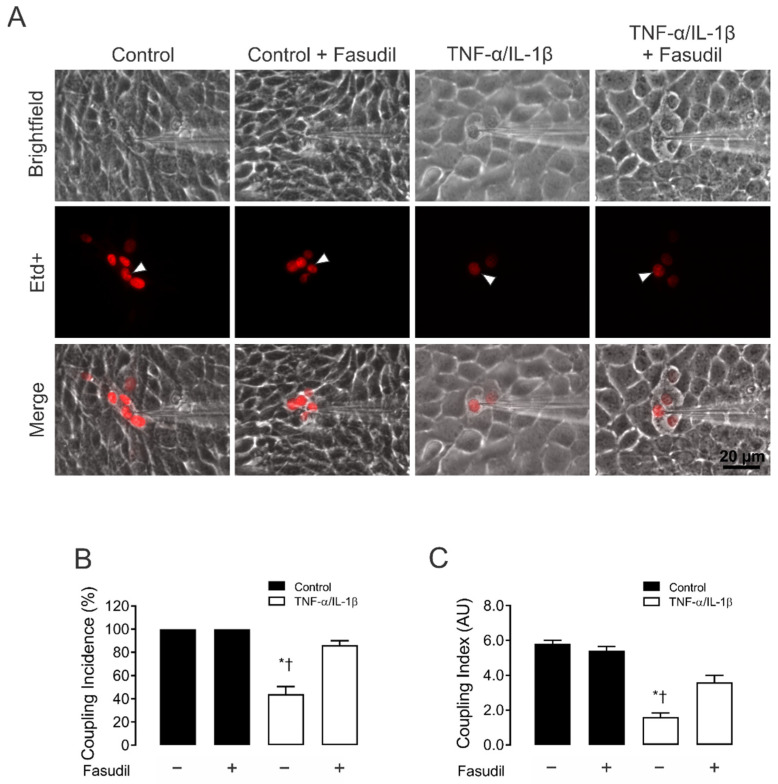
Fasudil prevents the TNF-α/IL-1β-induced reduction in gap junctional communication in MES-13 cells. (**A**) Gap junctional communication was examined using the microinjection of Etd^+^ (25 μM) in single cells (white arrows). After 2 min, the diffusion to neighboring cells was imaged. The top panels represent phase-contrast images, whereas the bottom panels show Etd^+^ fluorescence in cells under control conditions or upon different treatments. Scale bar = 20 μm. (**B**) Coupling incidence and (**C**) coupling index in confluent mesangial cells, under control conditions (black bars) or exposed to TNF-α/IL-1β (10 ng/mL) for 72 h (white bars). Fasudil (15 µM) was added together with TNF-α/IL-1β. Each bar represents the mean value ± SEM of 4 independent experiments. In each experiment, the dye was microinjected into at least 10 cells. Statistical significance * *p* < 0.05 vs. Ctrl; † *p* < 0.05 vs. TNF-α/IL-1β + Fasudil.

**Figure 3 ijms-23-10097-f003:**
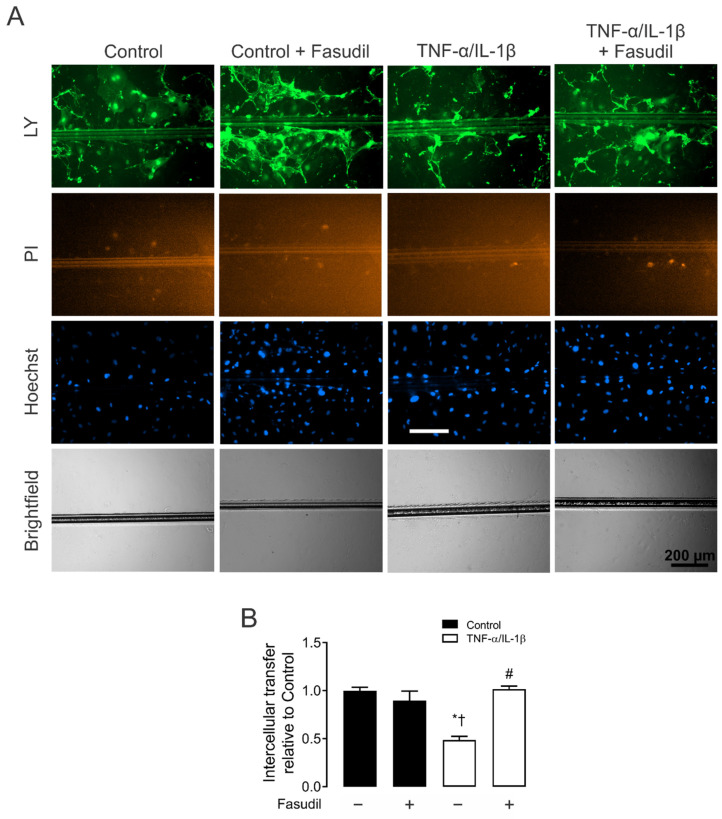
Fasudil prevents the TNF-α/IL-1β-induced decrease in primary mesangial cell coupling. (**A**) Representative fluorescence micrographs of SL/DT with Lucifer yellow (LY) by primary MCs under control conditions, 72 h after exposure to TNF-α/IL-1β (10 ng/mL) or treatment with Fasudil (15 µM). For each experiment are shown representative images of communicating cells stained with LY, dead cells along the cut stained with propidium iodide (PI), the total cells stained with Hoechst. Bar = 200 μm. (**B**) Averaged data normalized to control (black bars) of SL/DT with LY by primary MCs after 72 h of treatment with TNF-α/IL-1β (white bars) or in combination with Fasudil. Each bar represents the mean value ± SEM of 3 independent experiments. Statistical significance, * *p* < 0.05 vs. Ctrl; † *p* < 0.05 vs. Control + Fasudil; # *p* < 0.05 vs. TNF-α/IL-1β.

**Figure 4 ijms-23-10097-f004:**
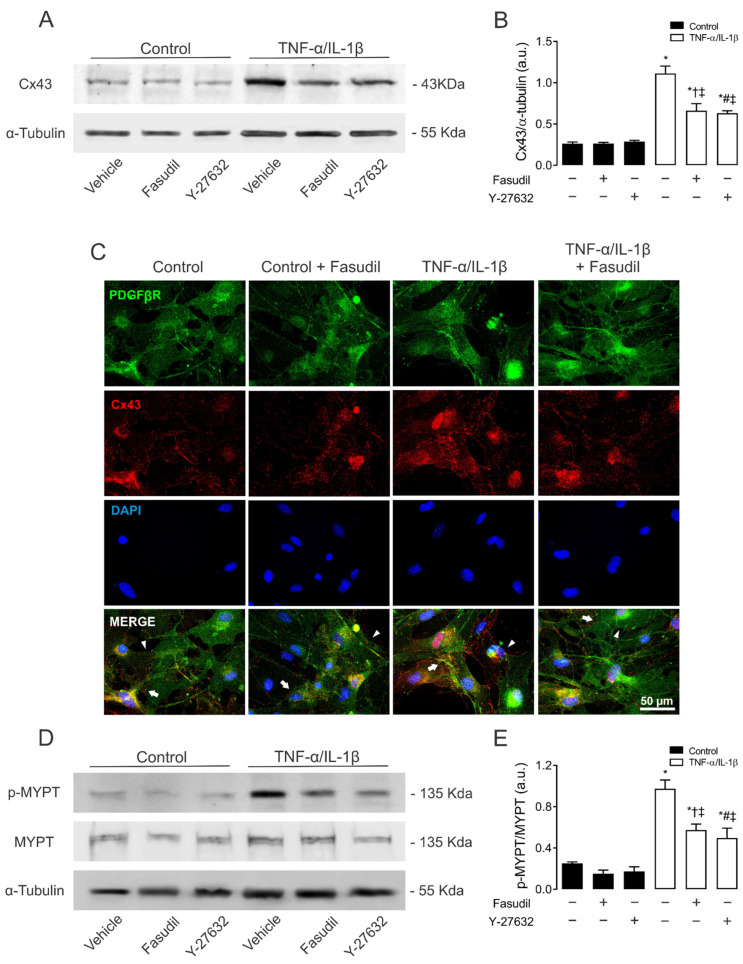
Blockade of ROCK prevents the increase in connexin43 and phosphorylated MYPT in mesangial cells induced by TNF-α/IL-1β. (**B**) Graphs show the relative amount of Cx43 and (**E**) phosphorylation of MYPT-1 determined by Western blotting analysis in MES-13 cells under control conditions (black bars) or exposed to TNF-α/IL-1β (white bars) for 72 h. (**C**) Fluorescence images depicting PDGFβR (green), Cx43 (red) and DAPI (blue) staining of primary MCs stimulated for 72 h with 10 ng/mL TNF-α/IL-1β. White arrows indicate GJ plaques at cell–cell interfaces, and white arrowheads indicate Cx43 HCs. Calibration bar: white = 50 μm. Fasudil (15 µM) or Y-27632 (15 µM) was added 24 h before harvesting the cells. Each bar represents the mean value ± SEM of ≥3 independent experiments. Statistical significance * *p* < 0.05 vs. Ctrl; † *p* < 0.05 vs. Control + Fasudil; # *p* < 0.05 vs. Control + Y-27632; ‡ *p* < 0.05 vs. TNF-α/IL-1β. Next to the graph, representative pictures of Cx43 (**A**), phosphorylated MYPT (p-MYPT), unphosphorylated MYPT positive bands (**D**), and its loading control (α-tubulin) are shown.

**Figure 5 ijms-23-10097-f005:**
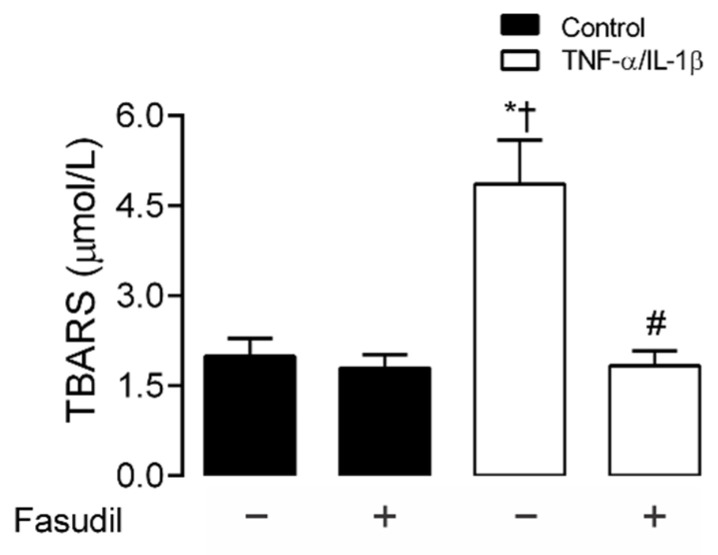
Blockade of ROCK prevents the increase in TBARS induced by TNF-α/IL-1β in primary mesangial cells. Graphs showing the amount of TBARS obtained from culture media of primary MCs under control conditions (black bars) or exposed to TNF-α/IL-1β (white bars) for 72 h. Fasudil (15 µM) was added 24 h before harvesting the cells. Each bar represents the mean value ± SEM of 4 independent experiments. Statistical significance, * *p* < 0.05 vs. Ctrl; † *p* < 0.05 vs. Control + Fasudil; # *p* < 0.05 vs. TNF-α/IL-1β.

**Figure 6 ijms-23-10097-f006:**
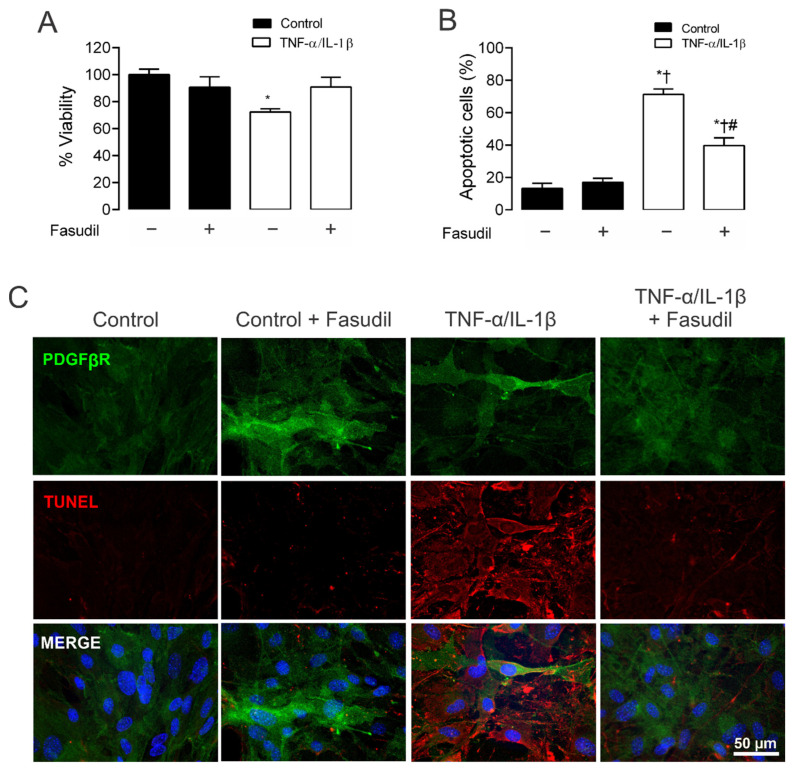
Blockade of ROCK prevents apoptosis, preserving cell viability in primary mesangial cells stimulated with TNF-α/IL-1β. Graphs showing the measurement of (**A**) cell viability and (**B**) % apoptotic cells in MCs under control conditions (black bars) or exposed to TNF-α/IL-1β (white bars) for 72 h. (**C**) Fluorescence images depicting PDGFβR (green), Tunel (red) and DAPI (blue) staining by primary MCs stimulated for 72 h with 10 ng/mL TNF-α/IL-1β. Calibration bar: white = 50 μm. Fasudil (15 µM) was added 24 h before harvesting the cells. Each bar represents the mean value ± SEM of 3 independent experiments. Statistical significance * *p* < 0.05 vs. Ctrl; † *p* < 0.05 vs. Ctrl + Fasudil # *p* < 0.05 vs. TNF-α/IL-1β.

**Figure 7 ijms-23-10097-f007:**
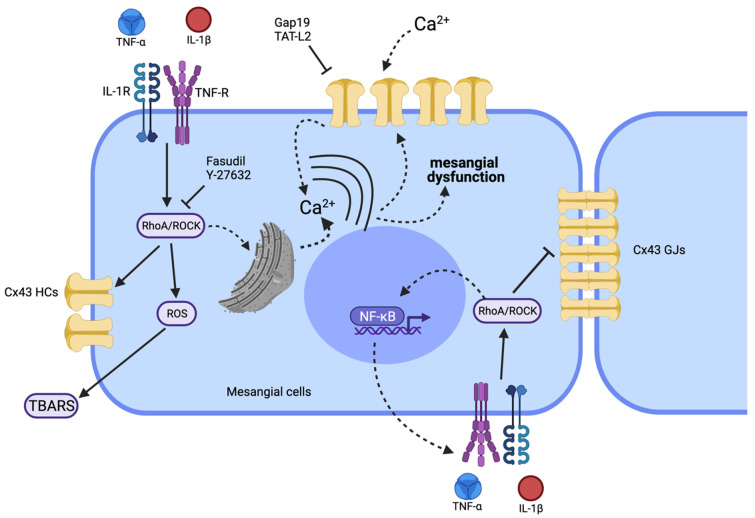
Schematic showing the possible signaling pathways involved in the TNF-α/IL-1β-mediated regulation of the functional state of Cx43 HCs and GJs in mesangial cells. High TNF-α/IL-1β concentrations could promote activation of a RhoA/ROCK-dependent pathway through TNF-α/IL-1β receptors (TNF-R and IL-1R) (continuous black arrows). Once the RhoA/ROCK-dependent pathway is activated, the formation of reactive oxygen species (ROS) takes place (generating TBARS) along with NF-κB-dependent production of proinflammatory cytokines such as TNF-α and IL-1β. The activated RhoA/ROCK-dependent pathway could increase the activity of Cx43 HCs, since Fasudil or Y-27632 (ROCK blockers) inhibit this response. The resulting increase in [Ca^2+^]_i_ will also activate Cx43 HCs, allowing the increase in Ca^2+^ influx. The latter results in the rise of [Ca^2+^]_i_ and further generation of ROS. The influx of Ca^2+^ establishes a positive feedback loop sensitive to different compounds such as the mimetic peptides Gap19 and TAT-L2, selective Cx43 HC blockers. This increase in the cellular activity caused by TNF-α/IL-1β, where the RhoA/ROCK pathway could be involved, also reduces cell–cell communication through GJs. Of note, alterations in [Ca^2+^]_i_ homeostasis mediated by Cx43 HCs or Panx1 HCs might affect diverse aspects of MC functions (e.g., morphology and proinflammatory profile). Discontinuous black arrows indicate cell responses identified in other systems, whereas straight black arrows denote responses identified in the present work. Elaborated in biorender.com.

## Data Availability

Not applicable.

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
