# Peer review of "TNF-α Plus IL-1β Induces Opposite Regulation of Cx43 Hemichannels and Gap Junctions in Mesangial Cells through a RhoA/ROCK-Dependent Pathway"

_ijms, 2022, doi:10.3390/ijms231710097_

Round 1
Reviewer 1 Report (Previous Reviewer 4)
Sir,
I have studied the re-submitted manuscript with particular interest. The authors have provided a very detailed point-by-point rebuttal letter, and I am grateful for that. It is evident that the authors honestly attempted to answer all my previous questions. And they did well. Additional experiments were performed and corroborated the hypothesis sufficiently.
Author Response
Sir,
I have studied the re-submitted manuscript with particular interest. The authors have provided a very detailed point-by-point rebuttal letter, and I am grateful for that. It is evident that the authors honestly attempted to answer all my previous questions. And they did well. Additional experiments were performed and corroborated the hypothesis sufficiently.
R: Thank you very much for your comment. It was a good job that could be modified from cell line to primary culture and interesting results could be obtained that are shown in this new version of the article.

Reviewer 2 Report (Previous Reviewer 3)
The manuscript has been significantly improved, although some minor revisions are required before acceptance. Please see comments below:
p.4 l.164 remove comma after AT2 receptors
p.5 l.255 for different time periods
p.6 l.272 What was the confluency of cells used for dye uptake? Report if coupled or single cells were used.
p.7 l.322 typo Immunofluorescence
p.7 l.326 What was the product code of the anti-Cx43 polyclonal antibody from SIGMA?
p.7 l.344/5 Report product codes of the anti-p-MYPT1 and anti-MYPT1 antibody
p.8 l.373 number of ≥15 cells per field
p.8 l.374 data were quantified by measuring fluorescence in five representative fields
p.8 l.402/3 No change in Etd+ uptake was observed
p.9 l.407 Supplementary Fig. 1 Figure label and figure legend missing
p.9 l.413 with sequences
p.9 l.414 intracellular loop domain of Cx43
p.11 l.446 (B) Fluorescence images
p.11 l.448/9 five independent experiments with four replicates each.
p.12 l. 477 remove “and”: we decided to explore whether TNF-α/IL-1β could affect the functional state of GJs in MES-13 cells by measuring the intercellular diffusion of...
p.14 l.517 after exposure to TNF-α/IL-1β
p.14 l.518 For each experiment are shown
p.14 l.519 dead cells along the cut stained with propidium iodide (PI)
p.14 l.520 Was DAPI used to stain total cells or Hoechst?
p.16 l.545 legend to Fig.4 E missing, update Fig. legend (see below)
p.16 l.546/7 (B) Graphs show the relative amount of Cx43 and (E) phosphorylation of MYPT-1 determined by western blotting
p.16 l.549 staining of primary MCs
p.17 l.555/6 phosphorylated MYPT (p-MYPT) and unphosphorylated MYPT positive bands (D), together with the loading control
p.17 l.581 the extracellular amount of TBARS was significantly lower
p.18 l.589 Fasudil (15 µM) was added
p.18 l.601/2 These data indicate that a RhoA/ROCK-dependent pathway increases OS in primary MCs which negatively affects cell viability.
p.19 l.615 value ± SEM of 3 independent experiments
p.20 l.655/6 Of note, blockade of RhoA/ROCK pathway also reduced the TNF-α/IL-1β-induced production of OS
p.21 l.670-2 Therefore, due to the multiple roles of MCs, they are considered a critical element in the origin and progression of various kidney diseases.
p.21 l.692ff Lillo et al. found that NO directly activates Cx43 HCs by S-nitrosylation of cysteine at position 271 (PMID: 31751316)
Author Response
The manuscript has been significantly improved, although some minor revisions are required before acceptance. Please see comments below:
p.4 l.164 remove comma after AT2 receptors
R: Thank you very much for your comment; it was corrected
p.5 l.255 for different time periods
R: Thank you very much for your comment; it was corrected
p.6 l.272 What was the confluency of cells used for dye uptake? Report if coupled or single cells were used.
R: Considering this assessment that you mention, the mesangial cells were used at 70% confluence. Now indicated in Material and Methods
p.7 l.322 typo Immunofluorescence
R: Thank you very much for your comment; it was corrected
p.7 l.326 What was the product code of the anti-Cx43 polyclonal antibody from SIGMA?
R: Thank you very much for your comment; the product code was #C6219 SIGMA. Now indicated in Material and Methods.
p.7 l.344/5 Report product codes of the anti-p-MYPT1 and anti-MYPT1 antibody
R: Thank you very much for your comment; the product codes were #ABS45 Merck Millipore for p-MYPT1 and #612164 BD Transduction Laboratories anti-MYPT1, respectively. This information is now indicated in Material and Methods sections.
p.8 l.373 number of ≥15 cells per field
R: Thank you very much for your comment; it was corrected
p.8 l.374 data were quantified by measuring fluorescence in five representative fields
R: Thank you very much for your comment; it was corrected
p.8 l.402/3 No change in Etd+ uptake was observed
R: Thank you very much for your comment; it was corrected
p.9 l.407 Supplementary Fig. 1 Figure label and figure legend missing
R: Thank you very much for your comment and in the new version the information of that figure had already been added.
p.9 l.413 with sequences
R: Thank you very much for your comment; it was corrected
p.9 l.414 intracellular loop domain of Cx43
R: Thank you very much for your comment; it was corrected
p.11 l.446 (B) Fluorescence images
R: Thank you very much for your comment; it was corrected
p.11 l.448/9 five independent experiments with four replicates each.
R: Thank you very much for your comment; it was corrected
p.12 l. 477 remove “and”: we decided to explore whether TNF-α/IL-1β could affect the functional state of GJs in MES-13 cells by measuring the intercellular diffusion of...
R: Thank you very much for your comment; it was corrected, and sentence was rewritten
p.14 l.517 after exposure to TNF-α/IL-1β
R: Thank you very much for your comment; it was corrected
p.14 l.518 For each experiment are shown
R: Thank you very much for your comment; it was corrected
p.14 l.519 dead cells along the cut stained with propidium iodide (PI)
R: Thank you very much for your comment; it was corrected
p.14 l.520 Was DAPI used to stain total cells or Hoechst?
R: Thank you very much for your comment and Hoechst was used. It was corrected
p.16 l.545 legend to Fig.4 E missing, update Fig. legend (see below)
R: Thank you very much for your comment; it was corrected, and sentences were rewritten
p.16 l.546/7 (B) Graphs show the relative amount of Cx43 and (E) phosphorylation of MYPT-1 determined by western blotting
R: Thank you very much for your comment; it was corrected, and sentences were rewritten
p.16 l.549 staining of primary MCs
R: Thank you very much for your comment; it was corrected
p.17 l.555/6 phosphorylated MYPT (p-MYPT) and unphosphorylated MYPT positive bands (D), together with the loading control
R: Thank you very much for your comment; it was corrected
p.17 l.581 the extracellular amount of TBARS was significantly lower
R: Thank you very much for your comment; it was corrected
p.18 l.589 Fasudil (15 µM) was added
R: Thank you very much for your comment; it was corrected
p.18 l.601/2 These data indicate that a RhoA/ROCK-dependent pathway increases OS in primary MCs which negatively affects cell viability.
R: Thank you very much for your comment; it was corrected
p.19 l.615 value ± SEM of 3 independent experiments
R: Thank you very much for your comment; it was corrected
p.20 l.655/6 Of note, blockade of RhoA/ROCK pathway also reduced the TNF-α/IL-1β-induced production of OS
R: Thank you very much for your comment; it was corrected
p.21 l.670-2 Therefore, due to the multiple roles of MCs, they are considered a critical element in the origin and progression of various kidney diseases.
R: Thank you very much for your comment; it was corrected
p.21 l.692ff Lillo et al. found that NO directly activates Cx43 HCs by S-nitrosylation of cysteine at position 271 (PMID: 31751316)
R: Thank you very much for your comment; it was corrected, and sentence was rewritten

This manuscript is a resubmission of an earlier submission. The following is a list of the peer review reports and author responses from that submission.
Round 1
Reviewer 1 Report
The study is purely additive and is a repeat of their former study which demonstrates similar concepts i.e the effect of Ang II on hemichannel activity. In this instance, the authors merely use stimuli downstream of Ang II which they have previously shown to be up regulated in response to Ang II treatment. The study utilises only immortalised mesangial cells so the impact is limited and in my opinion given the number of figures, the data not strong enough for a paper of increasing impact. I think the biggest issue i have is the complete lack of references within the field . The authors claim that we know nothing of how Cx HCs are linked to renal function and in doing so fail to cite many of the studies by key Pi's in the field including, Chadjichristos C and Hills CE. In addition, the references are incorrect with the authors citing review articles instead of the experimental studies , none of which are correctly referenced throughout. The study feels relatively basic, with minimal techniques and limited impact. The English is not good and in my opinion given the lack of novelty would be better suited for a lower impact journal.
Reviewer 2 Report
TNF-α plus IL-1β Induces opposite regulation of hemichannels and gap junctions in mesangial cells through a RhoA/ROCK-3 dependent pathway
It is very interesting and clinically relevant in vitro study aimed to reveal the mechanisms by which Cx43 hemichannels promotes an inflammation-induced injury of mesangial cells. Authors demonstrated that there is an opposite regulation of of Cx43 hemichannels and Cx43 gap junction channels by RhoA/ROCK-3-dependent pathway in response to inflammatory cytokines, TNF-α and IL-1β. Multiple methods were used to obtain convincing findings that were well demonstrated. Nevertheless, there are some issues to be addressed or to clarified prior publication.
Specific comments
Title: should specified that the regulation of Cx43 hemichannels and Cx43 gap junction channels were explored.
Abstract:
The first sentence should be revised because it makes impression that Cx43 expressed in the kidney is just for promoting inflammation somewhere. Second sentence is general, instead, it would be appreciated to specify conditions in which functional relationship between these membrane channels will be investigated.
Introduction:
Less is more, therefore, focusing on the most relevant information in the context of the study intention would help readers to know the purpose of the study. Paragraph at line 67 is out of scope of interest in the context of this study. Next paragraphs should also be profoundly shortened and references should be used instead to include general information about connexins. Some data can be used rather in discussion.
Taking into consideration of the published data, including authors previous work, the aim of the study should be clearly defined and it is not the best strategy to include the findings obtained in the study at the end.
Methods:
Line 168, intercellular coupling
To include the characteristic of the cell line MES-13 in respect to viability and intercellular as well as intra-extracellular communication would be appreciated. What is presumed proportion of Cx-HC and Cx43 GJ-channels in MES-13 cells?
Results:
To better catch the findings of the current study they should not be mixed with previous findings (plus references) that should be used in either the chapter Discussion or in association with the aim of the study in Introduction.
One would expect faster onset of the pro-inflammatory activity of the Cx43-HC in vivo conditions. Can you speculate about the timing in in vivo conditions?
Why the response of Cx43-HC to TNF-α/IL-1β did not occur earlier but only after 3 days, i.e. 72 hrs, of exposure to these inflammatory molecules?
In parallel, why the inhibitory effects of TNF-α/IL-1β on GJ-Cx43 channels did not occur earlier as well?
Is there some limitation of the model? How many cells die due to exposure to TNF-α/IL-1β?
It appears that the modulation of the function of both type of channels by TNF-α/IL-1β takes place in parallel, at least in this model. Can we expect dual influence at the same time in vivo as well?
Figure 4 shows that the overall protein level of Cx43 in MES-13 culture is increased due to TNF-α/IL-1β.
Is it possible to determine the level of Cx43 attributed to Cx43 hemichannels? If not, an increase of both GJ-Cx43 and Cx43-HC should be considered.
In this context it is interesting an observation that myocardial connexin-43 is upregulated six-weeks after acute cardiac injury in two rat models (Viczencova et a. 2017). One can speculate that upregulation of Cx43 in response to acute stress is self-defense mechanism to maintain intercellular communication when it is reduced. What do you think about in your model in which TBARS were markedly increased?
Despite the comprehensive Discussion there is luck of explanations to the points outlined above and related to the findings. Moreover, to extrapolate in vitro findings to in vivo conditions should be always with caution. Therefore, it is expected that authors will touch this issue and perhaps note some limitations of the study and/or suggest in vivo study based on current findings.
Reviewer 3 Report
Please find comments attached

Reviewer 4 Report
Sir,
I have recently reviewed the manuscript "TNF-α plus IL-1β Induces opposite regulation of hemichannels and gap junctions in mesangial cells through a RhoA/ROCK-dependent pathway" submitted by Claudia M. Lucero and co-workers to IJMS.
The authors decided to study two prominent cytokines - TNFa and IL1b - in mesangial cells of the kidney. Authors aimed to follow RhoA/Rock pathway signalling because this could be potentially targetted by available inhibitors. This signalling machinery could be of great importance in several inflammatory renal conditions. I believe that this topic is biologically interesting and possibly can have therapeutic implications in the future.
The introduction is well written. It is concise and provides sufficient insight even for a novice reader.
After reading The materials and methods, however, I have to immediately raise a critical point: all the experimental work was done on a single cell line, MES-13. The interpretation of such data is always dubious. Is it a unique feature in this cell line, or is it somewhat more general? The authors failed to present another biological replicate and/or control. Modelling of mesangial cells might be intricate (other cell lines or patient-derived isolates are not always available). However, I am sure that using various fibroblasts (of renal or extrarenal origin) can be an entirely sufficient surrogate in this case. Mesangial cells share a lot of features typical for fibroblasts ...
Related to the statistics: microinjection is a complicated procedure, I must sadly confirm that. Authors injected (line 281) at least 10 cells in certain experiments. The number seems to be low for the assumption of normal distribution (line 198). However, it is unclear which statistical method was selected for a particular experiment. The authors should write the statistical section more clearly, and also Figure description must contain these particular details. Also, the authors use multiple comparisons, and p-values should be corrected by sufficiently powerful procedure (e.g. Bonferroni) in such cases. The authors declared that "* p < 0.05 vs. Ctrl; # p < 0.05 vs. TNF-α/IL-1β." However, it leaves an impression (hopefully false) that they tried to avoid this. The authors must clarify this issue.
Authors (in Results) claim that "No changes in Etd+ uptake were observed in cells that were stimulated for shorter periods than 72 h or when TNF-α or IL-1β were added alone". It is a very interesting finding. Not unusual, combinations are always more powerful. However, I believe that timing is critical here, and it is somewhat surprising - it is rather long! Presumably, the "inflammatory" microenvironment represented by TNFa/IL1b shifted the phenotype of mesangial cells. Therefore it is not a direct consequence of cytokines in the medium, and it requires further analysis (e.g. IL-6/IL-8 are hot candidates for investigation). This must also be more carefully presented in the discussion.
Authors also claim (quote, line 269): "the same metabolic pathway that increases the activity of HCs seems to be involved in the reduction of cell-cell communication". I believe that this statement very nicely illustrated the proinflammatory "tuning" in their experimental design. This is also worthy of a more mechanistic explanation in the discussion. Of note, the metabolic activity of the untreated/treated system would also be interesting (even if you use simplistic, e.g. MTT or XTT based assays for this purpose). Related to Cx43 and its role in an inflammatory situation, it is not entirely new. I would ask the authors to deal with the older work of David L. Becker Group related to skin wound healing (see: https://doi.org/10.1371/journal.pone.0037374 ). I believe that this older work shares surprisingly a lot of common topics with the presented manuscript. I think the authors would benefit from another point of view.
To conclude, the manuscript brings some interesting data. However, I believe that certain above-listed aspects do not allow me (as a reviewer) to support it for acceptance now. I must honestly acknowledge the evident potential of this work. And I am also very keen to see the progress of this experimental work in the future. I would encourage the authors to strengthen their work and resubmit the paper soon.